# Immunotherapy as a Complement to Surgical Management of Hepatocellular Carcinoma

**DOI:** 10.3390/cancers16101852

**Published:** 2024-05-12

**Authors:** Susan J. Kim, Kaelyn C. Cummins, Allan Tsung

**Affiliations:** Department of Surgery, University of Virginia, Charlottesville, VA 22908, USA

**Keywords:** hepatocellular carcinoma, immunotherapy, surgery, locoregional treatment, neoadjuvant, adjuvant

## Abstract

**Simple Summary:**

Hepatocellular carcinoma (HCC) is the most common primary liver tumor in adults worldwide. Management of HCC has evolved substantially with the advent of immunotherapy. Despite these advances, evidence and formal treatment guidelines are limited to late stages of HCC. While early stages of HCC is currently managed with locoregional treatment, there is promising evidence for the use of concurrent immunotherapy. The aim of this review is to summarize existing evidence and identify ongoing trials regarding immunotherapy use in early-stage HCC amenable to locoregional treatment, as well as identify current gaps in knowledge.

**Abstract:**

Hepatocellular carcinoma (HCC) is the most common primary liver tumor in adults, and the fourth leading cause of cancer-related deaths worldwide. While surgical and ablative therapies remain the standard of care in early localized disease, late presentation with advanced stages of disease, impaired hepatic function, or local recurrence following surgical resection preclude operative management as the sole treatment modality in a subgroup of patients. As such, systemic therapies, namely immunotherapy, have become an integral part of the HCC treatment algorithm over the past decade. While agents, such as atezolizumab/bevacizumab, have well-established roles as first-line systemic therapy in intermediate- and advanced-stage HCC, the role of immunotherapy in disease amenable to surgical management continues to evolve. In this review, we will discuss the current evidence and aggregate impact of immunotherapy in the context of HCC amenable to surgical management, including its application in the neoadjuvant and adjuvant settings.

## 1. Introduction

Liver cancer is the seventh most common cancer, and the third leading cause of cancer-related deaths worldwide [1,2], with hepatocellular carcinoma (HCC) comprising 75% of primary liver cancers. The majority of HCC arises in the setting of cirrhosis, which has been associated with a variety of risk factors including viral hepatitides, alcohol use [3,4], and metabolic syndromes [5,6] including nonalcoholic fatty liver disease (NAFLD) [7,8,9,10,11]. Each etiologic subtype confers differences in time to diagnosis, severity, and disease course, ultimately leading to the variability seen in HCC-related mortality [12,13,14,15,16]. In addition to the variability of clinical features of HCC, the global incidence of disease is vascillating, given the unique geographic distribution of the known risk factors. For example, most cases due to endemic hepatitis infections arising in sub-Saharan Africa and eastern Asia [17], and these countries have seen declining incidence of disease since employing risk-reduction programs such as hepatitis B (HBV) vaccinations and aflatoxin reduction [18,19,20]. Conversely, Western populations continue to display steadily increasing rates of HCC, likely secondary to rising obesity and alcohol use [21,22,23]. In the US, a continued rise in incidence is expected through 2029, and anticipated to continue to disproportionately affect racial ethnic minorities, with the highest incidence in Hispanic and black patients at 4.7% and 4.3%, respectively [24].

Healthy liver parenchyma is a highly anti-inflammatory environment, allowing tolerance to benign foreign molecules such as food antigens [25]. This anti-inflammatory milieu is maintained by the interaction of antigen-presenting non-parenchymal Kupffer, hepatostellate, liver sinusoidal endothelial cells, and regulatory T cells (T_reg_) [26]. However, chronic inflammation as a result of hepatic disease can alter this balance, resulting in an environment supportive of tumorigenesis [27]. Tumor-associated macrophages, tumor-associated neutrophils, and myeloid-derived suppressor cells work in tandem to shift to and amplify an oncogenic phenotype and exert immunosuppressive effects, resulting in diminished natural killer (NK) cell and T_reg_ function, which ultimately lead to an environment suitable for HCC development [28,29,30]. While inflammation initiates oncogenesis, HCC cells perpetuate and maintain this pro-tumorigenic environment by the recruitment of dysfunctional immune cells and the upregulation of immune checkpoints [31]. Together, these changes form a highly immunogenic tumor with an intrinsic ability to evade the immune system. With this knowledge, immunotherapies have been explored as therapy options that may take advantage of the immunogenic nature of HCC and provide greater benefit with a more tolerable side effect profile than conventional cytotoxic chemotherapy. Since 2017, immune checkpoint inhibitors have been studied as monotherapies, dual therapies, and combination therapies in HCC. Findings from these studies have changed the landscape of HCC treatment, with atezolizumab and bevacizumab displacing sorafenib as first-line therapies in advanced disease [32], and introducing immunotherapy as the forefront in the management of HCC.

Less than 40% of patients presenting with HCC are stratified into earlier stages of disease, for which the standard of care is locoregional treatment (i.e., surgical resection, ablation, and chemoembolization) without systemic therapy. However, there are many factors that can prohibit surgical feasibility in these patients, such as unfavorable tumor anatomy or a high burden of comorbidities. Additionally, certain etiologies of HCC portend worse outcomes after locoregional management. A single center study demonstrated that post-transplant outcomes were more favorable in patients with NASH-related HCC compared to HCV-, HBV-, or alcoholic liver disease-related HCC [33]. Furthermore, certain HCC etiologies, particularly involving metabolic dysregulation, have been associated with poorer overall survival after curative locoregional therapies [34,35]. While the exact biologic mechanisms for this effect are unknown, there are suggestions that the distinct biochemical characteristic of each disease state may play a role in these outcome disparities. With this knowledge that a subset of patients may be poor candidates for the current standard of care, systemic therapies have been introduced as options for both primary and adjunctive treatment in earlier stages of disease. In this review, we will discuss the historic and current role of immunotherapy in the treatment of HCC, focusing on tumors amenable to locoregional intervention, as well as developing immunotherapeutic strategies.

## 2. Historic Use and Perspective on Immunotherapy in HCC

Historically, the treatment of HCC was categorized into locoregional treatment for tumors contained within the liver and systemic chemotherapy reserved for advanced disease. Advanced disease was initially treated with systemic chemotherapy, however no systemic chemotherapeutic agents demonstrated improved overall survival [36,37,38,39,40,41]. This prompted the 2008 Sorafenib Hepatocellular Carcinoma Assessment Randomized Protocol (SHARP) trial, which was able to demonstrate an overall survival benefit with the use of a tyrosine kinase inhibitor (TKI), sorafenib, in advanced HCC with preserved liver function [42], which has become the reference standard for the treatment of advanced HCC. After finding the overall survival benefit, the STORM trial was conducted in 2015 addressing the adjuvant timing of sorafenib. Patients with complete radiological response after resection or ablation of HCC received either sorafenib or a placebo, measuring for recurrence-free survival. There was no difference in median RFS between the groups, determining that sorafenib was not an effective intervention in the adjuvant setting [43]. Regardless, the results of overall survival benefit shifted systemic therapy strategies from cytotoxic chemotherapy to systemic tyrosine kinase inhibitors (TKI). Additional TKIs followed suit, with lenvatinib designated as a noninferior first-line therapy in 2017, regorafenib and cabozantinib as second-line therapies [44], and ramucirumab as an alternative to those who have failed other systemic therapies [45].

Following the finding of overall survival benefit with TKIs, investigation into immunotherapeutic agents began. Checkmate040, a phase I-II open-label, non-comparative, dose escalation and expansion trial published in 2017 demonstrated the tolerability of nivolumab monotherapy, although response rates remained low at 15–20% with no significant improvement in overall survival over the standard of care [46]. Subsequently, in the open-label phase III Checkmate495 trial, nivolumab alone did not demonstrate superiority in overall survival compared to sorafenib but did demonstrate some clinical activity with a favorable safety profile [47], establishing a promising foundation for future trials and utility in patients where TKIs were contraindicated. Later cohorts of Checkmate040 began examining combination therapy with nivolumab and ipilimumab among patients with advanced HCC who had previously underwent treatment with sorafenib. Astonishingly, an improvement in objective response rates by 30% was seen in the dual immunotherapy treatment arm, with a continued manageable safety profile [48,49,50]. This finding ultimately led to accelerated approval for this combination in 2020 for unresectable HCC previously treated with sorafenib. Currently, Checkmate 9DW (NCT04039607), a phase III study, is testing nivolumab/ipilimumab combination therapy in patients who had undergone prior therapy with lenvatinib, which may further support use of this combination.

Similarly, the KEYNOTE-224 trial in 2018, a nonrandomized open-label phase II trial examined pembrolizumab monotherapy in patients with Barcelona Clinic Liver Cancer (BCLC) stage B and C patients who were intolerant of or had disease progression on sorafenib therapy. The patients who received pembrolizumab demonstrated durable response and favorable progression-free survival (PFS) and overall survival (OS), with a maintained safety profile [51,52], throughout efficacy updates [53], as well as expansions in patients who had not undergone any prior systemic therapies [54]. Subsequent expansion demonstrated efficacy in unresectable HCC with preserved liver function, with OS comparable to nivolumab in Checkmate040.

Given evidence that combination therapy may provide added benefit, multiple additional trials were conducted to explore the effects of a variety of combined immunotherapeutic regimens. IMBrave150 demonstrated that atezolizumab and bevacizumab improved PFS and OS in advanced cancer compared to sorafenib alone, and this combination was approved as first-line therapy for advanced HCC [55] in 2020. In 2022, further advances were made in dual immunotherapy regimens, with the HIMALAYA study demonstrating that the STRIDE infusion regimen, consisting of a single priming dose of tremelimumab with regular interval doses of durvalumab, improved median overall survival in patients who had previously failed treatment with sorafenib. Furthermore, it demonstrated the durvalumab monotherapy was noninferior to sorafenib [56,57].

More recently, large multicenter trials have been conducted testing combination TKI-immunotherapy, but overall have not exhibited success in meeting pre-designated endpoints. The COSMIC-312 trial [58] evaluated cabozantinib with atezolizumab compared to sorafenib for advanced HCC and found no improvement in overall survival with dual therapy. Additionally, the LEAP-002 trial [59] tested Lenvatinib and pembrolizumab combination therapy and found that this combination also did not demonstrate improved overall survival or progression free survival compared to Lenvatinib monotherapy.

## 3. Current Recommendations for Immunotherapy in HCC

Expert guidelines for the management of HCC are detailed in the Barcelona Clinic Liver Cancer (BCLC) treatment recommendations [60], most recently updated in 2022. BCLC staging is based on tumor, liver, and patient characteristics, and subsequently stratifies each stage into potential treatment pathways (Figure 1). BCLC recommendations reserve systemic therapies primarily for BCLC C disease, with certain exceptions for BCLC B disease. Currently, atezolizumab–bevacizumab or durvalumab–tremelimumab are first-line recommendations for systemic therapy. Immunotherapeutic regimens are also second- and third-line recommendations for systemic therapy, replacing tyrosine kinase inhibitors as the standard of care. Additionally, the volume of clinical trials involving immunotherapeutic agents has substantially increased.

## 4. Evidence for Immunotherapy in Early/Intermediate Disease

Although there is well-established evidence regarding the efficacy and benefit of immunotherapy for HCC, these findings and subsequent guidelines are limited to addressing advanced disease. Currently, BCLC treatment recommendations for immunotherapy outside of advanced disease is limited to BLCL B disease when trans-arterial chemoembolization (TACE), trans-arterial radioembolization (TARE), or other locoregional therapies such as radiation therapy has failed or is infeasible; the utility of immunotherapy in very early (BCLC 0)-, early (BCLC A)-, and most intermediate (BCLC B)-stage disease is not well understood. While immunotherapy is neither recommended nor well understood in the earlier disease stages, there remain cases of tumor not amenable to locoregional treatment that systemic therapies may prove beneficial in. Particularly, neoadjuvant and adjuvant immunotherapy for downstaging and recurrence prevention respectively, in surgically amenable disease has been in the forefront of research, with promising early results.

### 4.1. Neoadjuvant Immunotherapy for Downstaging

With locoregional therapies being the mainstay of treatment in the early stages of disease, neoadjuvant immunotherapy has been explored as an adjunctive measure to augment the efficacy of locoregional treatment. Specifically, downstaging has emerged as a reliable method to reduce tumor burden to allow patients either to fall within acceptable Milan criteria for transplantation or to qualify for resection. A 2022 randomized open-label phase II trial tested perioperative nivolumab monotherapy versus nivolumab and ipilimumab in 27 patients with resectable HCC who had not had previous immunotherapy exposure [61]. A total of 13 patients received treatment with nivolumab monotherapy, while the remaining 14 patients received combination therapy with nivolumab and ipilimumab. A total of 20 of the 27 patients were able to undergo partial hepatectomy, meeting the safety and tolerability endpoints. Of the 20 patients who underwent resection, 6 achieved major pathological response, including 5 who achieved complete response. The group of complete responders was found to have a mean recurrence-free survival of 24.6 months. The 14 patients that did not achieve major pathological response were found to develop recurrence within the follow-up period.

ChiCTR1900023914, a single arm phase II trial explored lenvatinib and anti-PD-1 antibodies as conversion therapy for patients with unresectable intermediate to advanced HCC [62]. This trial studied the effect of systemic therapy in unresectable BCLC B and C HCC, with lenvatinib and anti-PD1 antibodies. Primary endpoint was conversion success by meeting resectability criteria determined by investigator assessment; secondary outcomes included objective response rate by mRECIST/RECIST criterion, RFS, OS, and safety. A total of 55.4% of participants were determined to successfully meet resectability criteria and proceeded to surgical resection. Among those patients, 38.1% demonstrated pathological complete response, and tumor histology was notable for intratumoral CD8+ T cell enrichment.

Ablative therapies and chemoembolization are thought to augment the natural immune response through local ischemia, releasing tumor neoantigens that prime the surrounding microenvironment to increase efficacy of systemic immunotherapeutics. Preclinical data have supported this hypothesis; Duffy et al. analyzed the immune responses of 32 patients after receiving a course of neoadjuvant tremelimumab followed by either radiofrequency or chemoablation, which demonstrated accumulation of intratumoral CD8+ T cells. When observing this phenomenon clinically, a phase I–III trial with tremelimumab and ablative therapies was conducted among 19 patients, 5 of whom demonstrated partial response [63]. In histopathological examination of those patients, responders had increased CD8+ infiltration in their tissue biopsies, correlating CD8+ T cell infiltration with improved clinical benefit. Additionally, in a study by Zhu et al., of 20 patients with intermediate-stage HCC that underwent neoadjuvant TACE and PD-1 inhibition with either camrelizumab or sintilimab as a bridge to surgery, 14 were successfully downstaged [64]. Among those who were successfully downstaged, an increased disease-free survival and overall survival were seen, though this finding was not statistically significant.

### 4.2. Adjuvant Immunotherapy

While neoadjuvant therapy is generally used as an approach to downstage and improve subsequent resectability, the goal of adjuvant therapy is to reduce recurrence rates and improve both disease-free and overall survival. Several solid organ cancers have demonstrated a benefit from adjuvant therapy; however, HCC is currently the only cancer without proven and recommended adjuvant therapy. Since the failure of the STORM trial to meet pre-designated end points, the utility of adjuvant therapy has been a topic of debate. Multiple therapies have been tested in the adjuvant setting with some promising results. Pre-existing therapies, such as adjuvant TACE [65,66,67,68,69] and hepatic artery infusion [70,71,72], have demonstrated some improvement in disease-free survival. Additionally, repurposed medications such as angiotensin receptor blockers [73] and antivirals [74,75,76] have demonstrated some survival benefit and increased time to recurrence for hepatitis-related HCC. Altogether, these findings have provided compelling evidence for the benefit of adjuvant therapy, with immunotherapy remaining at the forefront of research interest.

Mizukoshi et al. analyzed immune responses before and after radiofrequency ablation in 69 patients, which demonstrated enhanced T cell responses to HCC tumor-specific antigens at 24 weeks [77], providing pre-clinical evidence to support further research into adjuvant immunotherapy. To date, IMbrave050 is the only trial with positive results regarding immunotherapy in the adjuvant setting for early-stage disease. This phase 3 trial explored an atezolizumab–bevacizumab combination therapy and active surveillance after surgical resection, demonstrating statistically significant and clinically meaningful improvement in recurrence-free survival among patients in the treatment arm (HR 0.72) [78]. While there are no additional trials with published results, there are several current trials underway addressing immunotherapy in the adjuvant setting. The KEYNOTE-937 is a randomized double-blind phase 3 trial comparing pembrolizumab to placebo following surgical resections or ablations for curative intent, evaluating for recurrence risk reduction via complete radiologic response. Accrual has been completed, but formal data are pending [79]. Checkmate-9DX explores similar endpoints, however, evaluates nivolumab monotherapy in reducing recurrence in those at high risk after curative-intent surgical resection [80]. EMERALD-2 examines durvalumab and bevacizumab combination, monotherapies, and placebo in HCC with high risk of recurrence after resection or local ablation, hypothesizing that this population could have the greatest benefit with adjuvant therapy [81]. There are at least 12 ongoing trials examining immune checkpoint inhibition as adjuvant therapy in early disease.

Aside from immune checkpoint inhibitors, other modalities of immunotherapy have been tested in the adjuvant setting. Multiple clinical trials have demonstrated the overall favorable results of autologous lymphocyte transplantation. Takayama et al. demonstrated that infusion with adjuvant autologous lymphocytes activated with recombinant IL-2 and anti-CD3 decreased the rate of HCC recurrence by 18%, prolonged time to first recurrence (48% vs. 33% at 3 years, *p* = 0.008), and elongated time of recurrence-free survival, though no differences in overall survival were noted [82]. Similar findings were demonstrated by Lee et al.; the group conducted a phase III trial exploring whether adjuvant administration of autologous cytokine-induced killer cells (CIK) would prolong recurrence-free survival in HCC patients. They identified a significant increase in recurrence-free survival (median time 44.0 months vs. 30.0 months, HR 0.63, *p* = 0.01) and lower rates of all-cause and cancer-related deaths (HR 0.21, *p* = 0.008; HR 0.19, *p* = 0.02) in the immunotherapy treatment arm [83].

Vaccine therapy has recently gained popularity in oncologic research, particularly in the setting of immunogenic cancers. To date, trials have been limited to phase I or II designs, and the evidence for recurrence prevention when used in the adjuvant setting has been underwhelming. A phase II open-label single arm study conducted in Japan explored the use of a Glypican-3 peptide vaccine as an adjuvant therapy after resection or radiofrequency ablation. A total of 41 patients who had completed treatment within one year received a series of 10 vaccinations. The vaccination arm demonstrated recurrence rates of 28.6% and 39.5% at 1 and 2 years, respectively, compared to 39.5% and 54.5% recurrence in the non-vaccine case control arm. Although their intended primary endpoints were not met, this study demonstrated some efficacy and demonstrated a potential utility of GPC3 expression as a biomarker for future therapies [84]. Additionally, in a South Korean phase I/IIa study, 12 patients with HCC that had undergone primary treatment were administered dendritic cell (DC) vaccines pulsed with multiple tumor-associated antigens. The study measured adverse events, time to progression, and associated immune responses. A total of 9 out of 12 patients demonstrated no tumor recurrence for up to 24 weeks; and those patients additionally demonstrated stronger anti-tumor responses as measured by lymphocyte proliferation and IFN-gamma. The median time to progression was 36.6 months in the group receiving the DC therapy compared to 11.8 months in the non-therapy group, which is very promising [85]. Despite these studies not meeting their predetermined primary endpoints, they have been able to demonstrate the presence of some positive immunologic effect and tolerable safety profiles in vaccine therapy, which provides promise for future studies.

Finally, T cell therapy is also being explored as an adjuvant therapy in early HCC. It remains in the earlier stages of investigation, with limited data available. Currently, a single-center, single-arm, open-label study out of Beijing, China is evaluating the safety and clinical benefit of T cell receptor (TCR)-redirected T cell therapy in patients with HCC secondary to HBV. This study is following safety, overall response rate, and 5-year overall survival in patients who receive this therapy following hepatectomy or radiofrequency ablation. It is anticipated to be completed in June of 2024 (NCT03899415).

### 4.3. Ongoing Trials for Surgically Amenable HCC (Table 1 and Table 2)

## 5. Special Considerations—Liver Transplant

A specific subset of patients with HCC are those who undergo curative intent liver transplantation (LT). The use of oncologic immunotherapy in liver transplant candidates and recipients is not well studied. Often, transplanted patients are excluded from immunotherapeutic trials, therefore the limited data are largely derived from case reports.

A proposed use of immunotherapy in early-stage HCC is as a “bridging therapy” to prevent further disease progression while awaiting definitive transplantation. Currently, there are only two trials examining pre-transplantation immunotherapy. The PLENTY trial (NCT04425226) examines pembrolizumab and lenvatinib prior to LT, observing for RFS and ORR. Additionally, the Dulect2020-1 trial (NCT0443322) is evaluating safety and efficacy of durvalumab and lenvatinib in HCC before transplantation, observing PFS and RFS. Unfortunately, pre-transplant immunotherapy is also associated with risk. Current evidence suggests a washout period of 4–8 weeks between the last dose of immunotherapy and transplantation, given that transplantation within one half-life of nivolumab has been associated with acute rejection in 75% of cases [86].

The dreaded complication of an allograft rejection is of great concern, and a principal outcome measure in all immunotherapy trials with transplanted participants. Allograft rejection occurs in 36% of transplanted patients following immune checkpoint inhibitor (ICI) therapy, with rejection-related mortality occurring in 21% [87,88]. A small pilot evaluation of seven post-transplant patients receiving PD-1 inhibitor therapy for either HCC or melanoma demonstrated that two patients rejected their graft within 24 days [89]. A separate 2021 systematic review of 19 patients undergoing treatment with either nivolumab or pembrolizumab for recurrent HCC was evaluated for complications following treatment with ICI, 6 (32%) of whom experienced graft rejection [90]. In these studies, it was observed that generally patients further out from transplantation faced lower risk of rejection when treated with immunotherapy. Conflictingly, other small retrospective studies, particularly examining nivolumab-based therapies, have demonstrated no increased rate of rejection, even with ICI therapy delivered within one day of transplantation [91,92]. Other groups have postulated that immunotherapy use in transplant populations has increased risk of immune-related adverse events such as hepatitis and colitis [93] and have argued that the efficacy of immunotherapy is diminished in already immunosuppressed patients. The question of whether PD-L1 expression in the allograft contributes to development of graft failure in the setting of immunotherapy use has been raised, proposing that PD-L1 may contribute to the allograft’s inability to evade host immunologic response. In fact, there has been some histopathological evidence that the presence of PD-L1 expression in a graft prior to beginning ICI therapy is associated with higher risk of rejection [91] which may explain the allograft rejection identified thus far. If PD-L1 expression is related to this risk, then there is a question if non-PD1 immunotherapy may pose a decreased risk in this population.

A small risk of complications may be tolerated if a therapeutic efficacy was demonstrated, however study results are conflicted on that metric as well. Outside of “bridging therapy”, immunotherapy has been studied in the post-transplant setting to address malignancies. Transplanted patients are uniquely at increased risk of extrahepatic malignancies, particularly skin malignancies, with incidence rates of 1.3% within the first year of transplantation, and 18.8% within 20 years [94]. Additionally, the rate of recurrent HCC in patients who had undergone LT for this indication was approximately 10–15%, often early after transplantation [90]. A pilot evaluation of solid organ transplant recipients who received PD-1 inhibitors for HCC or melanoma showed that among patients who did not reject their graft (*n* = 5), only one patient achieved complete response while three showed progressive disease [88]. Furthermore, the 2021 systematic review evaluating 19 patients receiving nivolumab or pembrolizumab for recurrent HCC demonstrated an ORR of 11%, median PFS of 2.5 ± 1-month, median OS of 7.3 ± 2.7 months after treatment with ICI. Pembrolizumab was associated with higher rates of complete response, improved PFS, and improved OS compared with nivolumab [90]. Thus far, a clear therapeutic efficacy has not been established. Without a definitive association between immunotherapy use and therapeutic efficacy that outweighs the risk of rejection, the question of the prohibitive risk of immunotherapy in transplanted patients must continue to be raised. Nevertheless, these findings come from limited data, and further research may elucidate a benefit of immunotherapy use in this population.

## 6. Gaps in Knowledge and Future Directions

Significant progress has been made in determining the benefit and appropriate role of immunotherapy in HCC, and trials focusing on earlier stages of disease have demonstrated promising preliminary results thus far. However, several unanswered questions remain that warrant further investigation. Although earlier stages of disease are conventionally treated with locoregional intervention, patient comorbidities, underlying liver dysfunction and morphology, and poor functional status may confer prohibitive surgical risk in a patient with an otherwise anatomically resectable tumor. Additionally, while liver transplantation remains an option for the treatment of early-stage disease, available donors remain scarce. To date, there are no data supporting whether immunotherapy alone is non-inferior to locoregional treatment in resectable disease. There may be a benefit to exploring whether upfront immunotherapy is an efficacious treatment modality for patients unable to undergo locoregional treatment or transplantation.

In addition to determining if immunotherapy can be effective in early stages of disease, we need to explore why some patients have a robust response to immunotherapy, while others do not. More investigation can be performed to explore if this is related to oncogenic variation, patient tumor immune environmental factors, or if there is a traceable biomarker that can predict immunotherapeutic response. Clarification of the factors that influence patient response may change how we approach patient selection, management pathways, and risk stratification.

## 7. Conclusions

Management of HCC has rapidly changed with the introduction and continued evolution of immunotherapy. While there has been a great deal of advancement, the optimization of existing therapeutic regimens, discovery of new benefits of immunotherapy, and incorporation of these findings into the current treatment algorithm will be paramount for continued growth. Overcoming challenges related to immune-related adverse effects and the use of immunotherapy in particular subpopulations, such as liver transplant recipients, may expand the subset of patients who would benefit from this therapy. With the ongoing and promising research in this field, the outlook for immunotherapy use in earlier stages of disease remains promising as an adjunct to locoregional care of HCC.

## Figures and Tables

**Figure 1 cancers-16-01852-f001:**
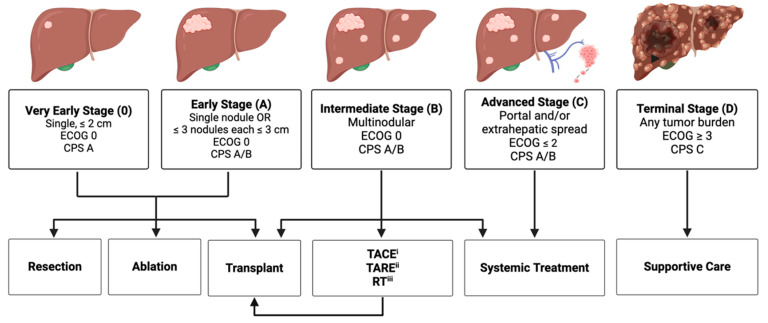
Abbreviated Barcelona Clinic Liver Cancer (BCLC) treatment algorithm. Created with www.BioRender.com (accessed on 4 March 2024). (i) Transarterial chemoembolization. (ii) Transarterial radioembolization. (iii) Radiation Therapy.

**Table 1 cancers-16-01852-t001:** Neoadjuvant Trials.

Trial ID	Population	Treatment Arm	Primary Endpoints	Secondary Endpoints
NCT03682276 (PRIME HCC)	Early-stage HCC	Nivolumab + ipilimumab	Delay to surgery. Safety and tolerability.	ORR PRR
NCT03299946	Borderline resectable HCC	Cabolizumab + nivolumab	Completion of treatment and proceeding to surgery. AE.	R0 resection CR MPR ORR OS DFS
NCT04658147	Technically resectable HCC	Nivolumab ± relatlimab	Treatment completion. proceeding to surgery.	AE R0 resection pCR/MPR ORR OS DFS
NCT03337841 (AURORA) *	Resectable HCC with high recurrence risk	Pembrolizumab	1-year RFS	RFS OS ORR Tumor markers AE
NCT04721132	Resectable HCC	Atezolizumab + bevacizumab	pCR AE	ORR DOR RFS OS
NCT05185531 (Notable-HCC)	Resectable HCC	SBRT + tislelizumab	Delay to surgery ORR pCR AE	DFS OS
NCT03510871	Potentially resectable HCC with high recurrence risk	Nivolumab + ipilimumab	Treatment response by RECIST	-
NCT05471674	Borderline resectable HCC	Nivolumab	PRR	RFS OS Short-term surgery outcomes AE
NCT03916627	Resectable HCC	Cemiplimab	Significant tumor necrosis	Delay to surgery Event free survival DFS ORR OS AE Change in tumor infiltrating CD8 density
NCT03867370 *	Technically resectable HCC	Lenvatinib Toripalimab	PRR	ORR R0 resection Time to operation PFS OS AE
NCT04850040	Locally advanced, potentially resectable, ruptured, adjacent organ invasion	Camrelizumab, apatinib mesylate, oxaliplatin	MPR	ORR 1-year RFS DFS AE
NCT04615143 (TALENT)	Resectable recurrent HCC	Lenvatinib, tislelizumab	DFS	ORR ISAE MPR
NCT05194293	Potentially resectable high-risk tumor T1b, T2, T3a	Regorafenib and durvalumab	16-week ORR	Proceed to surgery AE OS RFS PCR
NCT04888546	Resectable HCC with high recurrence risk	TQB2450 + anlotinib	pCR ORR	PFS OS ISAE
NCT04224480	Technically resectable	Pembrolizumab	2-year recurrence	Intratumoral Ki67 T cells
NCT05389527 (NeoLeap-HCC)	Technically resectable	Lenvatinib + pembrolizumab+	MPR	PCR ORR R0 resection rate DFS OS AE
NCT04930315 (CAPT) *	Technically resectable BCLC B/C	Camrelizumab + apatinib	1-year recurrence rate	OS RFS R0 resection MPR PCR Resection rate AE
NCT05185739 (PRIMER-1)	HCC with solitary tumor	Lenvatinib, pembrolizumab	MPR	% viable tumor cells at resection RRR RFS Delay to surgery 30-day surgical complications Treatment completion AE
NCT04954339 (DYNAmic) *	Potentially resectable BCLC B/C HCC	Atezolizumab, bevacizumab	PCR Tumor immunophenotype	Treatment completion R0 resection AE PFS RRR RFS

ORR (objective response rate); PRR (pathologic response rate); CR (complete response); MPR (major pathologic response); OS (overall survival); DFS (disease-free survival); AE (adverse events, i.e., toxicities, adverse events, and complications); RFS (recurrence-free survival); DOR (duration of response); ISAE (immune related serious adverse event); PCR (pathologic complete response); TTR (time to recurrence); TTLR (time to local recurrence); RRR (radiological response rate). * Joint neoadjuvant/adjuvant trials.

**Table 2 cancers-16-01852-t002:** Adjuvant Trials.

Trial (NCT)	Population	Treatment Arm	Primary Endpoint (s)	Secondary Endpoint (s)
NCT03337841 * (AURORA)	Resectable HCC with high recurrence risk	Pembrolizumab	1-year RFS	RFS OS ORR Tumor markers AE
NCT03383458 (Checkmate9DX)	BCLC 0/A HCC with high recurrence risk	Nivolumab	RFS	OS TTR
NCT04102098 (IMBrave050)	BCLC 0/A HCC with high recurrence risk	Atezolizumab + bevacizumab	RFS	OS RFS TTR OS Time to EHS/macrovascular invasion RFS in PD-L1-high subgroup AE
NCT03847428 (EMERALD-2)	BCLC 0/A HCC with high recurrence risk	Durvalumab vs. Durvalumab-bevacizumab	RFS	RFS OS TTR PFS
NCT03867084 (KEYNOTE-937)	BCLC 0/A HCC with CR after resection or ablation	Pembrolizumab	RFS OS	AE Treatment termination QOL change
NCT03859128 (JUPITER-04)	BCLC 0/A HCC	Toripalimab	RFS	RFS RFS 12/24 months TTR TTLR OS 12/24 months AE
NCT04981665	BCLC 0/A HCC	TACE with sequential tislelizumab	2-year RFS	RFS TTR OS 1-year RFS 1/2 year OS AEs
NCT04682210	BLCL 0/A HCC	Sintilimab + bevacizumab	RFS	OS RFS 12/24 months OS 24/36 months TTR AEs
NCT03867370 *	Technically resectable HCC	Lenvatinib Toripalimab	PRR	ORR R0 resection Time to surgery PFS OS AE
NCT04930315 (CAPT) *	Technically resectable BCLC B/C	Camrelizumab, apatinib	1-year RFS	OS RFS R0 resection MPR PCR Resection rate AE
NCT04954339 (DYNAmic) *	Potentially resectable BCLC B/C HCC	Atezolizumab, bevacizumab	PCR Tumor immunophenotype	Treatment completion R0 resection AE PFS RRR RFS

ORR (objective response rate); PRR (pathologic response rate); CR (complete response); MPR (major pathologic response); OS (overall survival); DFS (disease-free survival); AE (adverse events, i.e., toxicities, adverse events, and complications); RFS (recurrence-free survival); PCR (pathologic complete response); TTR (time to recurrence); TTLR (time to local recurrence); RRR (radiological response rate); QOL (quality of life). * Joint neoadjuvant/adjuvant trials.

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
