# Peer review of "Immunotherapy as a Complement to Surgical Management of Hepatocellular Carcinoma"

_cancers, 2024, doi:10.3390/cancers16101852_

Round 1

Reviewer 1 Report

Comments and Suggestions for Authors

Thi sis a review of immunotherapy in HCC

1) In the first paragraph discussing incidence, newer article (PMID: 38398075) can be cited. Also, racial difference in projected incidence can be mentioned

2) Line 84: It is mentioned "Following the introduction of TKIs as a superior systemic therapy to cytotoxic chemotherapy,". To the best of my knowledge, TKIs have not been studies in comparison to cytotoxic chemotherapy in HCC. Please amend

3) section 4.2: Preliminary results of Emerald-2 were presented at GI ASCO2024 and should be included

4) Figure 1: consider adding TARE

5) Please mentioning some failed studies of combining IO + TKI including len + pem, atezo + cabo etc 

Author Response

1) In the first paragraph discussing incidence, newer article (PMID: 38398075) can be cited. Also, racial difference in projected incidence can be mentioned

Thank you for the feedback and the updated reference. The new citation has been included in the revised version of the manuscript as citation #1. Additionally, racial differences in incidence has also been added to our section 1 to help provide a more complete and inclusive understanding of this disease process.

2) Line 84: It is mentioned "Following the introduction of TKIs as a superior systemic therapy to cytotoxic chemotherapy,". To the best of my knowledge, TKIs have not been studies in comparison to cytotoxic chemotherapy in HCC. Please amend

Thank you for identifying this flaw in verbiage. The intent was the convey that cytotoxic systemic therapeutics preceded sorafenib, and that those strategies were ineffective. The reviewer is correct in mentioning there were no head-on trials. As a result, we have revised this section of the review and have included the references for prior trials assessing cytotoxic chemotherapy and its lack of survival benefit prior to introducing TKI therapy.

3) section 4.2: Preliminary results of Emerald-2 were presented at GI ASCO2024 and should be included

Thank you for the update on these very important results. On further investigation, the preliminary results presented at GI ASCO2024 were from the Emerald-1 study, which focuses on embolization eligible unresectable HCC, with no outcome measures regarding downstaging. As the purpose of this review was to only investigate therapies for HCC that is amenable to surgical intervention or has the potential to be downstaged to resectability, this trial seems outside of the scope of this review.

4) Figure 1: consider adding TARE

Thank you for identifying this oversight. TARE was included in Figure 1 as a treatment modality for BCLC B disease.

5) Please mentioning some failed studies of combining IO + TKI including len + pem, atezo + cabo etc

Thank you for this suggestion. Both the COSMIC-312 and LEAP-002 trials were added to our section regarding the historic uses of immunotherapy in HCC. This addition definitely rounds out and provides a more complete overview regarding the history of immunotherapy in this disease.

Reviewer 2 Report

Comments and Suggestions for Authors

In your center(University of Virginia in Charlottesville) among the patients who have cirrhosis have you studied immunotherapy in an attempt to prevent hepatocellular carcinoma? Among patients with hepatocellular carcinoma have you studied xenotransplantation? Are you doing any studies with stem cells?

Regardless your paper will certainly be of interest to caregivers, basic scientists and pharmaceutical companies.

Author Response

In your center(University of Virginia in Charlottesville) among the patients who have cirrhosis have you studied immunotherapy in an attempt to prevent hepatocellular carcinoma? Among patients with hepatocellular carcinoma have you studied xenotransplantation? Are you doing any studies with stem cells?

Regardless your paper will certainly be of interest to caregivers, basic scientists and pharmaceutical companies.

Thank you for the very interesting questions as well as the feedback regarding those stakeholders that may be interested in our review. Regarding the institution-specific questions, our center does not have any ongoing trials regarding immunotherapeutic prophylaxis in patients with cirrhosis or xenotransplantation for patients with HCC. Our institution does have one ongoing clinical trial regarding stem cell transplantation in those with hematologic malignancies, however no investigation in the Department of Surgery or the Division of Gastroenterology. To my knowledge, we additionally do not have any groups studying these topics on a basic or translational level either.

Reviewer 3 Report

Comments and Suggestions for Authors

Very interesting and nicely written review on a cutting-edge topic. Since the failure of the STORM trial, the impact of adjuvant therapies after radical treatments (surgery or ablation) of HCC is matter of debate. The authors should comment more on this aspect mentioning quickly also the other treatments tested (for example cite the paper PMID: 25974743)

The authors should comment also on the potential impact of the underlying etiology of the liver disease on the clinical course of these patients

Author Response

Very interesting and nicely written review on a cutting-edge topic. Since the failure of the STORM trial, the impact of adjuvant therapies after radical treatments (surgery or ablation) of HCC is matter of debate. The authors should comment more on this aspect mentioning quickly also the other treatments tested (for example cite the paper PMID: 25974743)

The authors should comment also on the potential impact of the underlying etiology of the liver disease on the clinical course of these patients

Thank you for the kind commentary and the feedback for how to improve our review. Given the importance of the STORM trial and the questions that arose regarding adjuvant therapy following that study, we included more commentary to frame our thoughts in the adjuvant immunotherapy section (4.2). Regarding other treatments that were tested in the adjuvant setting, this review is narrowly focused on immunotherapy, therefore the addition of other tested treatments such as ACEi/Sartans may be outside of the scope of this review. However, it would be interesting to explore the complete scope of tested and available treatments for HCC outside of this review.

Round 2

Reviewer 3 Report

Comments and Suggestions for Authors

None of my previous points were addressed by the authors. The paper still needs a major revision in this form (see the points raised in the first stage of revision and completely ignored in this version)